

# Recently-adopted foraging strategies constrain early chick development in a coastal breeding gull

Alejandro Sotillo[1,2], Jan M. Baert[1,3], Wendt Müller[3], Eric W.M. Stienen[4], Amadeu M.V.M. Soares[2] and Luc Lens[1]

[1] Department of Biology, Terrestrial Ecology Unit, Ghent University, Ghent, Belgium
[2] Department of Biology and CESAM, University of Aveiro, Aveiro, Portugal
[3] Department of Biology, Behavioural Ecology and Ecophysiology Group, University of Antwerp, Antwerp, Belgium
[4] Research Institute for Nature and Forest (INBO), Brussels, Belgium

Corresponding author
Alejandro Sotillo,
alejandro.sotillogonzales@ugent.be

## ABSTRACT

Human-mediated food sources offer possibilities for novel foraging strategies by opportunistic species. Yet, relative costs and benefits of alternative foraging strategies vary with the abundance, accessibility, predictability and nutritional value of anthropogenic food sources. The extent to which such strategies may ultimately alter fitness, can have important consequences for long-term population dynamics. Here, we studied the relationships between parental diet and early development in free-ranging, cross-fostered chicks and in captive-held, hand-raised chicks of Lesser Black-backed Gulls (*Larus fuscus*) breeding along the Belgian coast. This traditionally marine and intertidal foraging species is now increasingly taking advantage of human activities by foraging on terrestrial food sources in agricultural and urban environments. In accordance with such behavior, the proportion of terrestrial food in the diet of free-ranging chicks ranged between 4% and 80%, and consistent stable isotope signatures between age classes indicated that this variation was mainly due to between-parent variation in feeding strategies. A stronger terrestrial food signature in free-ranging chicks corresponded with slower chick development. However, no consistent differences in chick development were found when contrasting terrestrial and marine diets were provided *ad libitum* to hand-raised chicks. Results of this study hence suggest that terrestrial diets may lower reproductive success due to limitations in food quantity, rather than quality. Recent foraging niche expansion toward terrestrial resources may thus constitute a suboptimal alternative strategy to marine foraging for breeding Lesser Black-backed Gulls during the chick-rearing period.

## INTRODUCTION

Human activities globally provide a growing amount of food subsidies such as household waste in cities, landfills, or marine fishing discards to free-ranging populations of animals (*Oro et al., 2013*; *Plaza & Lambertucci, 2017*; *Real et al., 2017*). Opportunistic feeders,

i.e., species that vary their diet with local food availability, are at the center of ecosystem responses to such subsidies as they can potentially switch to novel resources (e.g., *Tornberg, Mönkkönen & Pahkala, 1999*; *Sorace & Gustin, 2009*). In opportunistic species with wide home ranges, the abundance and diversity of anthropogenic food resources often implies that individuals may choose amongst an expanded range of potential diets (e.g., *Duhem et al., 2003*; *Moss et al., 2016*; *Navarro et al., 2010*; *Yoda et al., 2012*). This increase in ecological opportunity may then result in dietary niche variation within populations (*Bolnick et al., 2007*; *Navarro et al., 2017*). The abundance, accessibility (i.e., energetic costs associated with foraging), predictability, and nutritional value of the different anthropogenic food sources mediate the adaptive value of alternative diets (*Bicknell et al., 2013*; *Oro et al., 2013*). Given that anthropogenic food subsidies are characteristically prone to unpredictable variations in abundance driven by changes in human behavior (e.g., *Oro, Bosch & Ruiz, 1995*, *Oro et al., 2004*, *Steigerwald et al., 2015*), evaluating the impact of different anthropogenic diets on reproductive success may help predict the consequences of alterations in the abundance of a type of subsidy, such as a ban on discards or the closure of open air landfills (*Bicknell et al., 2013*; *Real et al., 2017*).

Many large gull species of the genus *Larus* make increasing use of human-dominated (i.e., urban and agricultural) terrestrial habitats for both feeding and breeding, albeit to a variable degree between and within breeding colonies (*Garthe et al., 2016*; *Matos et al., 2018*; *Mendes et al., 2018*; *Moreno et al., 2010*; *Osterback et al., 2015*; *Shaffer et al., 2017*). Within single colonies, some breeders specialize on particular marine or terrestrial food sources, whereas others consistently adopt more generalist strategies (*Camphuysen et al., 2015*; *Van den Bosch et al., 2019*; *Van Donk et al., 2017*). While intra-population niche partitioning in gulls has been previously linked to sex, age and personality (*Navarro et al., 2010*; *Navarro et al., 2017*), reported effects on chick development of terrestrial vs. marine diets are highly heterogeneous among studies. Different species show individual benefits from the exploitation of garbage (*Weiser & Powell, 2010*) or fish (*Annett & Pierotti, 1999*), but differences are also found between studies of the same species. For instance, *Hunt (1972)* described a positive relationship between reliance on garbage and breeding performance in Herring Gulls (*Larus argentatus*), *Pierotti & Annett (1991)* found Herring Gulls specializing in intertidal foraging to perform best, and *Van Donk et al. (2017)* claim that the exploitation of discards and garbage results in better breeding performance than that of intertidal organisms, while they found no differences between different degrees of specialization. In contrast, *Van den Bosch et al. (2019)* found resource specialization to positively impact chick growth in Herring Gulls exploiting intertidal and terrestrial foraging habitats. Benefits of newly adopted foraging strategies hence appear strongly context dependent, that is, to vary among individual traits and with environmental conditions.

Over the last decade, populations of Lesser Black-backed Gulls (*Larus fuscus*) have expanded their foraging niche toward more terrestrial diets (*Bicknell et al., 2013*). This process may consist of niche shifts driven by an overall decrease in marine food resources due to a decline in the availability of fishery discards (*Votier et al., 2004*; *Zeller & Pauly, 2005*), or represent a niche expansion following local cultural evolution (sensu *Danchin et*

*al., 2004*) due to increased ecological opportunity inland (e.g., *Moss et al., 2016*; *Newsome et al., 2015*). However, in gull populations exploiting a range of resources, chicks are most often fed a fish-based diet (e.g., *Alonso et al., 2015*; *Garthe et al., 1999*; *Hillström, Kilpi & Lindström, 1994*; *Isaksson et al., 2016*; *Skórka & Wójcik, 2008*). This suggests that terrestrial diets may have negative effects on chick rearing in gulls. If more terrestrial chick diets would indeed be found to result in lower reproductive performance than predominantly marine diets, niche diversification toward the exploitation of terrestrial food sources may present fitness costs.

We here tested for the potential effects of chick diet composition on chick growth and condition. To disentangle effects of diet composition from those of other environmental factors affecting individuals in natural populations, such as variation in food availability, we report on a two-year field study on cross-fostered and synchronized free-ranging chicks integrated with an experimental dietary study on hand-raised chicks in nearby outdoor aviary facilities. Diets of free-ranging chicks were determined by means of carbon and nitrogen stable isotope analysis of feather samples, while hand-raised chicks were provided with either a pure terrestrial or marine diet *ad libitum*. Body condition and growth were based on repeated measures until 30 days of age for both groups of chicks. Earlier studies showed that growth rates and body mass during chick development are positively correlated with future survival rates (*Lindström, 1999*; *Braasch, Schauroth & Becker, 2009*; *Bosman, Stienen & Lens, 2016*) and hence can act as proxy for fitness costs of parental foraging strategies. We expect a larger terrestrial diet component to result in poorer growth and condition in free-ranging chicks. These differences may be driven by food composition if are also observed in hand-raised chicks. Otherwise, the effects of diet on growth and condition of free-ranging chicks may reflect differences in the amounts of food provided.

## MATERIALS & METHODS

### Field study

We studied the development of free-ranging chicks in a mixed breeding colony of Lesser Black-backed Gulls and Herring Gulls (*Larus argentatus*) at the outer port of Zeebrugge (Belgium, 51°20′53″N 3°10′20″E). The colony counted 1458 Lesser Black-backed Gull pairs in 2016 and 1326 pairs in 2017 (*Stienen et al., 2017*; *Stienen et al., 2018*). Lesser Black-backed Gulls exhibit a limited sexual dimorphism (males being on average larger than females but with a large overlap) and show bi-parental care. The species lays two or three eggs during May and June, of which the first two are largest (*Verboven et al., 2005*). At the onset of breeding in 2016 and 2017, during egg laying, 26 and six Lesser Black-backed Gull nests, respectively, were haphazardly selected for monitoring and enclosed with chicken wire. Sample size was reduced in 2017 to avoid further disturbance of the colony after a year with high nest predator pressure. Nests were visited every first, third and fifth weekday, until all remaining chicks had reached an age of 30 days after the hatching date, thus covering the periods from 27 May to 19 July 2016 and from 7 June to 14 July 2017.

On the estimated day of hatching, eggs of monitored nests were substituted by 2 first- or second-laid eggs of equal developmental stage (= pipping eggs), each originating from

a separate nest. This resulted in a standardized clutch size of 2 and ensured hatching synchrony, thus preventing survival differences due to hatching asynchrony within and among clutches and ruling out effects of parental genetic quality on chick development. Upon hatching, nestlings were individually marked with colored tape, and down feathers were collected for molecular sexing. During each visit, chick body mass (to the nearest g) and head length (to the nearest mm) were measured. At 30 days of age, the right innermost primary feather (P1) of each monitored chick was plucked and stored for isotope analysis. In total, 15 female and 22 male chicks were sampled in 2016, and 5 females and 5 males were sampled in 2017.

## Aviary experiment

To avoid unnecessary disturbance of the Zeebrugge colony, Lesser Black-backed Gull eggs for the aviary experiment were collected from a nearby colony in the port of Ostend, Belgium (51°13′15″N2°56′27″E) on 29 May 2016. We consistently collected one of the two largest eggs (i.e., first or second laid egg) from haphazardly selected 3-egg clutches. Chicks hatching from larger eggs have a higher chance of successfully fledging at 30–40 days after hatching (*Del Hoyo et al., 2018*). To avoid laying date effects and ensure synchronized hatching, only pipping eggs were sampled. All collected eggs were placed in an incubator (temperature 37.5°C, humidity 62%) in a nearby aviary facility of Ghent University hosted in the Ostend Bird Rescue Center (VOC Oostende) until they were fully hatched. Of 34 collected eggs, 32 hatched between 30 May and 4 June 2016, 20 hatchlings (10 females and 10 males) were assigned to the current study. At hatching, chicks were randomly assigned to a diet of either ground whole adult chicken (*Gallus gallus domesticus*) or ground whole Sprat (*Sprattus sprattus*), with 10 chicks per diet treatment. These food types were chosen based on their macronutrient composition, which was representative of the food items found in regurgitates of chicks at the colony of Zeebrugge (see Table A1). We compensated for the loss of vitamins C and B1 due to prolonged deep-freeze storage by adding them to the thawed food. Food items were thawed at room temperature 6 to 8 h before feeding the chicks. On the first day after hatching, food was offered at five different times using tweezers. Afterwards, and during the remaining duration of the experiment, food was available *ad libitum* on a plate, and refreshed every 3 h between 9 AM and 9 PM. Chicks were measured every 5 days from the day of hatching until 40 days of age. We measured body mass (to the nearest g), total head length (head and bill, to the nearest 0.1 mm) and length of the external part of the P1 feather (measured to the nearest mm with a digital caliper). At 40 days of age, the P1 feather of all 20 chicks was plucked and stored for stable isotope analysis.

## Sexing

All field and aviary chicks were molecularly sexed using DNA samples extracted from down feathers. A polymerase chain reaction (PCR) was performed using *Fridolfsson & Ellegren (1999)* 2550F/2718R primers, and sex was subsequently determined by electrophoresis of these samples.
## Diet assessment

We estimated the terrestrial versus marine component in the diet of field chicks by means of carbon and nitrogen stable isotope analysis of feather samples, a technique that has proven efficient in assigning the proportion of assimilated diet components to a restricted number of sources in gulls, with marine food being characterized by higher values of $\delta^{13}$C and $\delta^{15}$N than terrestrial food (*Moreno et al., 2010*; *Steenweg, Ronconi & Leonard, 2011*; *Weiser & Powell, 2011*). First, we assessed the local diet of Lesser Black-backed Gulls based on an analysis of pellets and regurgitations of individuals breeding at Zeebrugge (Research Institute for Nature and Forest, 2006–2017; See Fig. SA1), supplemented by literature data (*Camphuysen, 2011*; *Garthe et al., 2016*). Based on this information, we collected and analyzed three samples of locally sourced swimming crabs (Subfam. *Polybiinae*), chicken meat, fish (Cod *Gadus morhua*, Sole *Solea solea*, Plaice *Pleuronectes platessa*), earthworms (Fam. *Lumbricidae*) and fried potatoes (*Solanum tuberosum*). All food samples were freeze dried, ground and subjected to accelerated solvent lipid extraction as described in *Bodin et al. (2009)*. Second, we cut P1 feathers of free-ranging chicks into 3 sections, each corresponding to a 10-day period. The length of feather sections in field chicks was estimated as the average length of P1 feathers in aviary chicks at 10, 20 and 30 days after hatching (values in Fig. SA1). Only feather vellum was used in the analysis of stable isotope ratios. All feather sections were cleaned for 5 min in an ultrasonic bath, left 12 h in a 2:1 chloroform –methanol wash, and oven-dried at 50 °C for 24 h. After this, food and feather samples were finely cut and placed in tin cups. Third, isotopic ratios were obtained by mass spectrometry at the Department of Applied Analytical and Physical Chemistry of Ghent University. Isotope ratios are reported in per mil (‰) using standard delta notation:

$$\delta = \left[ \left( \frac{Rsample}{Rstandard} \right) - 1 \right] \times 1000$$

where R represents the 13C/12C or 15N/14N ratio. Standards were Vienna Pee Dee Belemnite for carbon or air $N_2$ for nitrogen, respectively. Fourth, stable isotope signatures in feathers were corrected for tissue fractionation by means of trophic enrichment factors (TEFs). TEFs were calculated for carbon and nitrogen ($\delta^{13}$C and $\delta^{15}$N) between fish and chicken fed to aviary chicks, and the P1 feathers of these chicks, as in *Hobson & Clark (1992)*, following the formula: $\delta X = \delta a - \delta d$, where $\delta a =$ stable isotope composition of feather vellum tissue and $\delta d =$ stable isotope composition of the diet. TEF for a given isotope was then estimated as the average of individual $\delta X$ values for that isotope (see Table SA2 for the obtained values).

Next, the proportion of terrestrial food in the chicks' diet was estimated based on these carbon and nitrogen isotopic ratios and TEFs using a Bayesian stable-isotope mixing model (package MixSIAR, (*Stock et al., 2018*; *R Core Team, 2018*). Models were fitted using the Markov-chain Monte-Carlo algorithm, simulating 3 chains over 1000000 time steps. Model convergence was assessed by means of the Gelman–Rubin (*Gelman & Rubin, 1992*) and *Geweke (1992)* diagnostics.

## Data analysis

The estimated proportion of terrestrial food in the diet of field chicks was compared between three age periods (0–10, 10–20, and 20–30 days after hatching) and between sexes by means of a beta regression with identity link (*Ferrari & Cribari-Neto, 2004*) using R package betareg (*Cribari-Neto & Zeileis, 2010*). Significance of effects of the tested variables and their interaction were assessed with an analysis of deviance using Chi-squared tests. Following *Camphuysen (2013)*, a 3-parameter logistic growth curve was fitted to the body mass and total head length of each chick, from both field and aviary, approximating by least-squares the parameters of the logistic function:

$$y = \frac{a}{1 + be^{-kt}}$$

where $y$ is the body mass (g) or head length (mm), a is the corresponding upper asymptote, b the body mass or head length at the point of inflection, k is the growth rate ($days^{-1}$) and t is the number of days since hatching. The obtained upper asymptote and growth rate values were analysed by multiple regression with sex and the estimated proportion of terrestrial food in diet (field chicks) or diet treatment (aviary chicks), as well as their interactions, as explanatory variables. Significance of effects of the explanatory variables and their interactions were assessed by means of F-tests. Body condition was approximated using residual body mass, i.e., the residual values of a linear regression of body mass on total head length (*Reist, 1985*). Residual body mass values were averaged per individual within each age period. These were analysed by means of linear mixed effects models, using R package nlme (*Pinheiro et al., 2018*), with sex, age period and the proportion of terrestrial food in diet (field chicks) or the diet treatment (aviary chicks), and all their possible two- and three-way interactions, as explanatory variables. Chick identity was included as a random intercept.

Models were fitted using the Restricted Maximum Likelihood (REML) estimation to reduce bias in parameter estimations (*Harville, 1977*). Model residual normality and homoscedasticity were assessed respectively by means of a Shapiro–Wilk normality test (*Shapiro & Wilk, 1965*) and Breusch-Pagan test (*Breusch & Pagan, 1979*). Significance levels of all tests were set at 5%. Only significant interactions were retained, while main effects were always retained to avoid parameter estimation bias (*Whittingham et al., 2006*).

## Ethical approval

All applicable international, national, and institutional guidelines for the care and use of animals were followed during the aviary experiment (Ghent University Ethical Committee, project EC number 2015-017) as well as in the field study (University of Antwerp Ethical Committee for Animal Experiments, project EC number 2013-73). Additionally, all procedures performed in the aviary were in accordance with the regulations of the institution at which the studies were conducted (VOC Oostende).

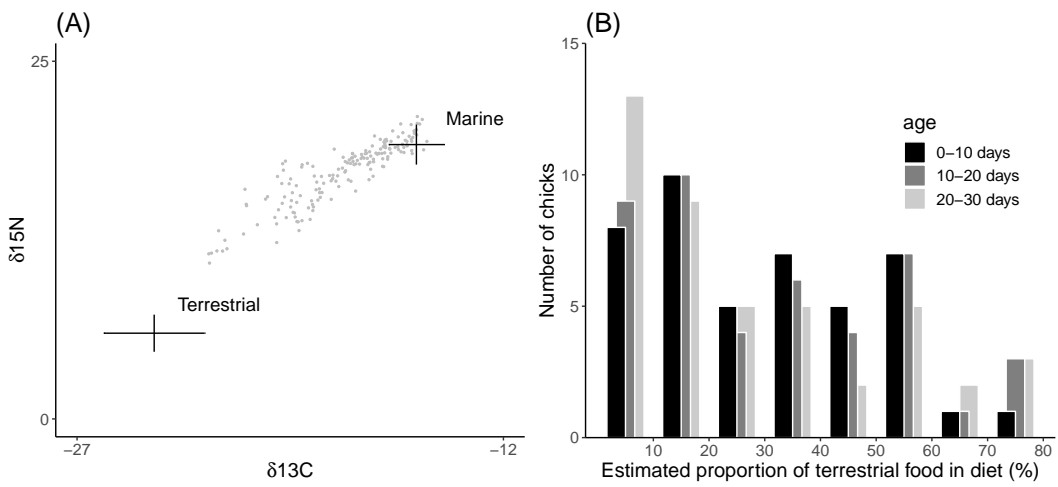

**Figure 1** **Distribution of diet compositions inferred from field chick feather samples.** (A) Isospace for values of $\delta^{15}$N and $\delta^{13}$C in *Lesser Black-backed* Gull feather samples relative to average ($\pm$1SD) values of food sources. Stable isotope ratios in food sources are corrected using trophic enrichment factors. (B) Frequency histogram for the estimated proportion of terrestrial food in chick diet.

**Table 1** **Beta regression model of the proportion of terrestrial food in diet of free ranging chicks in relation to sex and age period.**

|  | Estimated marginal mean proportion ± s.e | $\chi^2$ | d.f. | *p*-value |
|---|---|---|---|---|
| **Sex** |  | 2.06 | 1 | 0.151 |
| Female | 0.33 ± 0.03 |  |  |  |
| Male | 0.28 ± 0.02 |  |  |  |
| **Age** |  | 0.18 | 2 | 0.916 |
| 0–10 days | 0.31 ± 0.03 |  |  |  |
| 10–20 days | 0.31 ± 0.03 |  |  |  |
| 20–30 days | 0.29 ± 0.03 |  |  |  |

## RESULTS

### Variation in diet composition of field chicks

Estimated proportions of terrestrial food in the chick diet ranged from 4% to 80% (Fig. 1). The estimated proportions were concentrated at the more marine values (mean 30%, median 24% terrestrial component), implying that most field chicks were raised on a predominantly marine diet. The proportion of terrestrial food in the chick diet did not vary significantly in relation to chick age or sex (Table 1).

### Effects of diet composition on chick development

Asymptotic body mass of both field and aviary chicks was significantly related to their diet composition (Table 2). In field chicks, asymptotic body mass was negatively related to the proportion of terrestrial food (Fig. 2A), while aviary chicks reached a higher body mass when raised on a terrestrial diet (Fig. 2B). Males attained a higher body mass than females

**Table 2  Regression models of asymptotic size and growth rates of body mass (g) and head length (mm) in relation to sex and the proportion of terrestrial food in diet (field) or diet treatment (aviary).** Non-significant interactions were removed from each model.

| | Parameter | Variable | | Estimated marginal mean or coefficient ± s.e | F-statistic | d.f. | *p*-value |
|---|---|---|---|---|---|---|---|
| **Body mass** | | | | | | | |
| Field | Asymptote (g) | Proportion Terrestrial Food | | $-287.92 \pm 60.13$ | 24.01 | 1,42 | <0.001 |
| | | Sex | Female | $684.06 \pm 20.97$ | 45.74 | 1,42 | <0.001 |
| | | | Male | $859.29 \pm 15.21$ | | | |
| | Growth rate (days$^{-1}$) | Proportion Terrestrial Food | | $-0.20 \pm 0.95$ | 0.05 | 1,42 | 0.819 |
| | | Sex | Female | $5.61 \pm 0.33$ | 1.77 | 1,42 | 0.190 |
| | | | Male | $6.16 \pm 0.24$ | | | |
| Aviary | Asymptote (g) | Diet | Fish | $852.85 \pm 19.65$ | 11.72 | 1,17 | 0.003 |
| | | | Chicken | $947.98 \pm 19.65$ | | | |
| | | Sex | Female | $809.41 \pm 19.65$ | 42.9 | 1,17 | <0.001 |
| | | | Male | $991.42 \pm 19.65$ | | | |
| | Growth rate (days$^{-1}$) | Diet | Fish | $5.84 \pm 0.22$ | 10.60 | 1,17 | 0.005 |
| | | | Chicken | $6.86 \pm 0.22$ | | | |
| | | Sex | Female | $6.06 \pm 0.22$ | 3.52 | 1,17 | 0.078 |
| | | | Male | $6.64 \pm 0.22$ | | | |
| **Head length** | | | | | | | |
| Field | Asymptote (mm) | Proportion Terrestrial Food | | $-12.39 \pm 3.52$ | 13.32 | 1,42 | <0.001 |
| | | Sex | Female | $107.42 \pm 1.23$ | 63.76 | 1,42 | <0.001 |
| | | | Male | $119.53 \pm 0.89$ | | | |
| | Growth rate (days$^{-1}$) | Prop. Terr. Food | | $0.10 \pm 1.56$ | 0.004 | 1,42 | 0.95 |
| | | Sex | Female | $12.20 \pm 0.54$ | 4.00 | 1,42 | 0.052 |
| | | | Male | $13.55 \pm 0.40$ | | | |
| Aviary | Asymptote (mm) | Diet | Fish | $124.38 \pm 1.79$ | 0.003 | 1,17 | 0.955 |
| | | | Chicken | $124.53 \pm 1.79$ | | | |
| | | Sex | Female | $118.40 \pm 1.79$ | 22.88 | 1,17 | <0.001 |
| | | | Male | $130.50 \pm 1.79$ | | | |
| | Growth rate (days$^{-1}$) | Diet | Fish | $14.82 \pm 0.47$ | 1.01 | 1,17 | 0.330 |
| | | | Chicken | $15.48 \pm 0.47$ | | | |
| | | Sex | Female | $14.68 \pm 0.47$ | 1.99 | 1,17 | 0.177 |
| | | | Male | $15.61 \pm 0.47$ | | | |

in both field and aviary chicks, with similar effect sizes in both environments. Growth rates in field chicks (Fig. 2C) did not vary with diet or sex. In contrast, aviary chicks gained body mass faster when raised on a terrestrial diet (Fig. 2D). Male aviary chicks showed a trend toward faster body mass gain than females, which was however not found to be statistically significant (Table 2).

Asymptotic head length of field chicks was inversely related to the proportion of terrestrial food in their diet, while in aviary chicks, diet treatment had no effect (Table 2). Male chicks attained a larger head length than females in both the field and aviary (Figs. 3A, 3B). Growth rates for head length in field and aviary chicks were not significantly related to

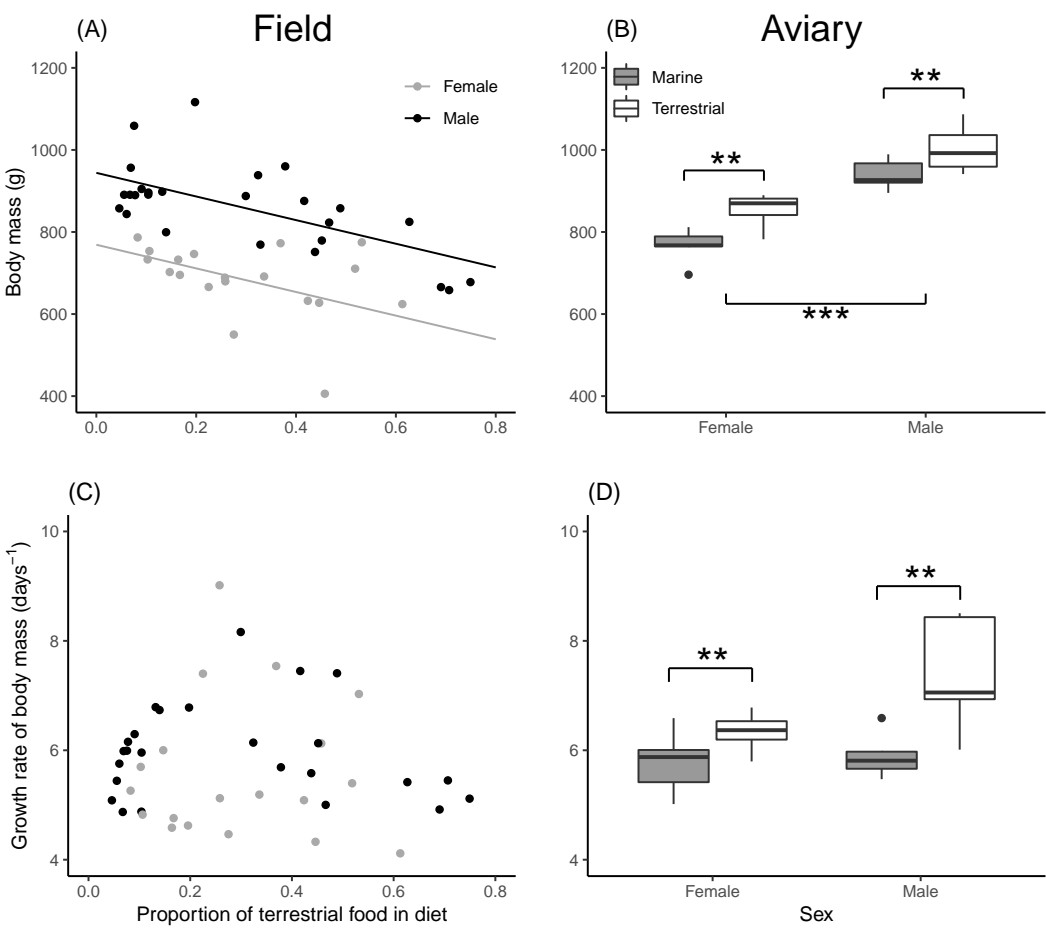

**Figure 2 Asymptote and growth rate of body mass in relation to diet composition.** (A) Estimated asymptotic body mass (g) of male and female field chicks plotted in relation to the estimated proportion of terrestrial food in the diet. Regression lines are plotted for significant relationships. (B) Box plot of the asymptotic body mass (g) in aviary chicks, separated by sex and dietary treatment. (C) Estimated growth rate (days$^{-1}$) plotted against the estimated proportion of terrestrial food in diet of field chicks, separated by *sex*. (D) Box plot of the estimated growth rate (days$^{-1}$) of body mass in aviary chicks, separated by sex and treatment. Boxes correspond to median, first and third quartile, and whiskers to 1.5 times the interquartile range outlier points. ***, $P < 0.001$; **, $P < 0.01$.

their diet (Table 2), and sex differences in growth rates of head length were not statistically significant in the field (Fig. 3C) or in the aviary (Fig. 3D).

## Effects of diet composition on body condition

For field chicks, the retained model for residual body mass contained an interaction between the proportion of terrestrial food in the diet and growth period (Table 3). During the first 10 days of growth (Fig. 4A), residual body mass did not vary with the proportion of terrestrial food, while in the second and third 10-day period (Figs. 4B, 4C), it decreased with increasing proportion of terrestrial food. In aviary chicks, males showed higher residual body mass than females (Figs. 4D, 4E, 4F), but no evidence was found for diet or age effects.

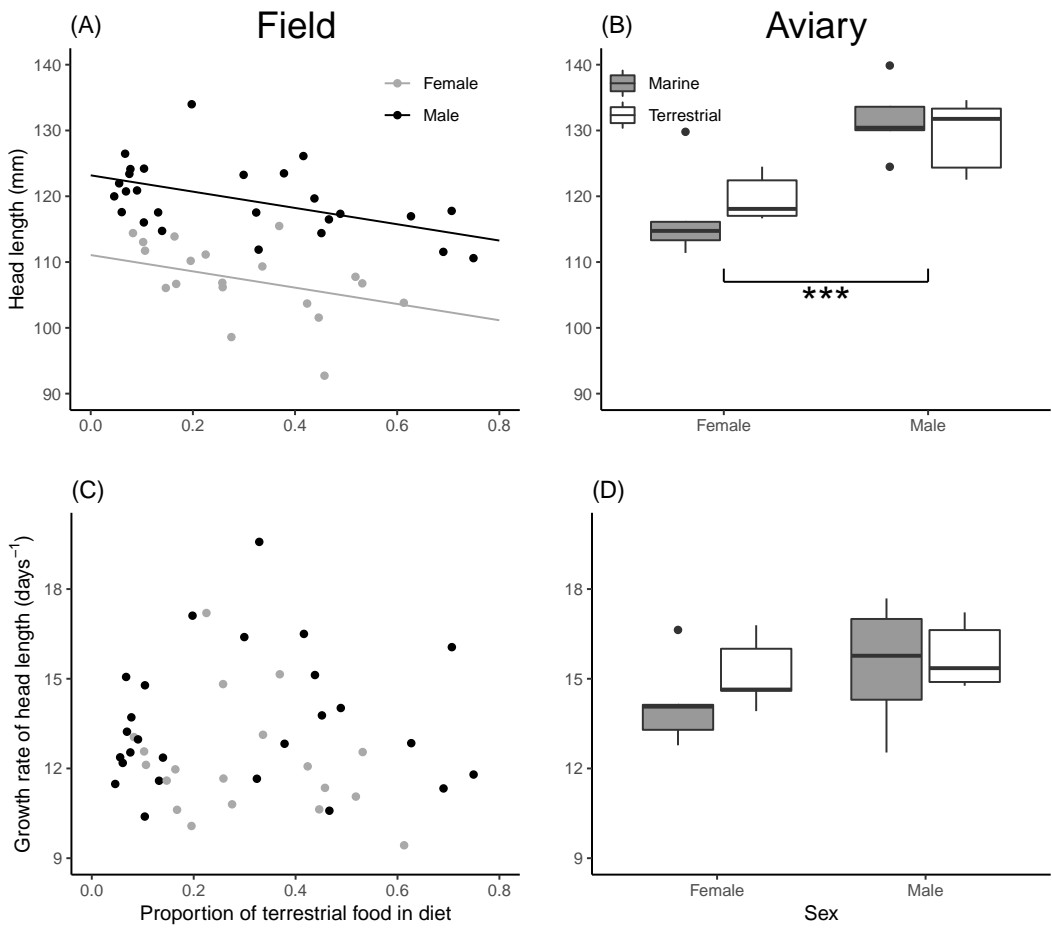

**Figure 3  Asymptote and growth rate of head length in relation to diet composition.** (A) Estimated asymptotic head length (mm) of male and female field chicks plotted in relation to the estimated proportion of terrestrial food in the diet. Regression lines are plotted for significant relationships. (B) Box plot of the asymptotic head length (mm) in aviary chicks, separated by sex and dietary treatment. (C) Estimated growth rate (days$^{-1}$) plotted against the estimated proportion of terrestrial food in diet of field chicks, separated by *sex*. (D) Box plot of the estimated growth rate (days$^{-1}$) of head length in aviary chicks, separated by sex and treatment. Boxes correspond to median, first and third quartile, and whiskers to 1.5 times the interquartile range outliers. ***, $P < 0.001$.

## DISCUSSION

Lesser Black-backed Gull chicks in the study colony of Zeebrugge showed a predominantly marine food signature. No cases of chicks fed solely on terrestrial food sources were detected, whereas an almost completely marine diet was frequent in our sample. Stable isotope signatures of individual chicks remained fairly constant throughout the rearing period, suggesting that the observed variation in diet composition was mainly due to consistent between-parent variation in feeding strategies, rather than temporary changes in food availability. Whereas a stronger terrestrial food signature corresponded with slower chick development under field conditions, no clear differences in chick skeletal development between terrestrial and marine diets occurred when food was provided *ad*

**Table 3 Linear mixed effects models of the residual body mass (g) of field and aviary chicks in relation to sex and the proportion of terrestrial food in diet (field) or diet treatment (aviary).** Non-significant interactions were removed from each model.

| | Estimated marginal mean or coefficient $\pm$ s.e | F-statistic | d.f. | p-value |
|---|---|---|---|---|
| **Field** | | | | |
| Proportion Terrestrial Food | | 0.36 | 1,57 | 0.165 |
| Sex | | 0.007 | 1,43 | 0.952 |
| Female | $1.23 \pm 5.52$ | | | |
| Male | $0.65 \pm 4.52$ | | | |
| Age | | 0.95 | 2,86 | 0.378 |
| 0–10 days | $5.10 \pm 4.86$ | | | |
| 10–20 days | $0.19 \pm 4.86$ | | | |
| 20–30 days | $-2.47 \pm 4.86$ | | | |
| Proportion Terrestrial Food x Age | | 3.15 | 2,88 | 0.047 |
| 0–10 days | $32.03 \pm 24.52$ | | | |
| 10–20 days | $-58.37 \pm 27.83^{*}$ | | | |
| 20–30 days | $-65.73 \pm 27.95^{*}$ | | | |
| **Aviary** | | | | |
| Diet | | 0.33 | 1,17 | 0.571 |
| Fish | $-4.70 \pm 3.72$ | | | |
| Chicken | $-1.67 \pm 3.72$ | | | |
| Sex | | 8.95 | 1,17 | 0.008 |
| Female | $-11.05 \pm 3.72$ | | | |
| Male | $4.68 \pm 3.72$ | | | |
| Age | | 0.1 | 2,38 | 0.379 |
| 0–10 days | $-6.06 \pm 3.39$ | | | |
| 10–20 days | $-2.55 \pm 3.39$ | | | |
| 20–30 days | $-0.95 \pm 3.39$ | | | |

**Notes.**
$^{*}p < 0.05$.

libitum under controlled aviary conditions. A larger asymptotic body mass, as well as a somewhat faster growth rate for this trait, were found for aviary chicks when fed on a terrestrial diet, but this did not result in a larger residual body mass, which would have been indicative of differences in the energy reserves of the individuals.

Higher trophic level diets, reflected by higher $\delta^{15}$N values (Ambrose & Deniro, 1986), have previously been linked to improved chick condition and higher breeding success in seabirds (e.g., (Bukacinska, Bukacinski & Spaans, 1996; Janssen et al., 2011; Ronconi et al., 2014; Van Donk et al., 2017). Here, we did not find an intrinsic difference between marine and terrestrial diets when provided ad libitum. Instead, dietary effects on chick development only became apparent under field conditions, and may thus relate to differences in cost-benefit ratios between marine and terrestrial foraging strategies. Food resource partitioning among individuals within a colony has often been ascribed to competitive differences in relation to body size, albeit mainly driven by differences between males and females in sexually dimorphic species (Camphuysen et al., 2015; Monaghan, Coulson

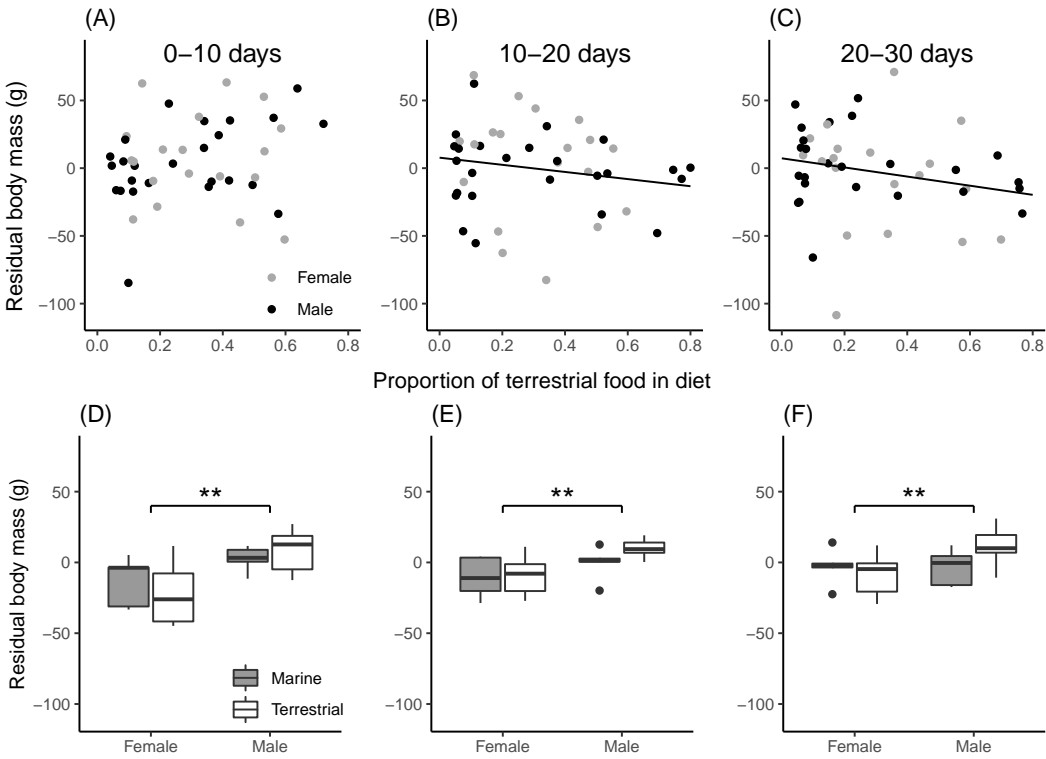

**Figure 4 Chick residual body mass in relation to diet composition.** (A–C): Residual body mass (g) of field chicks plotted in relation to the estimated proportion of terrestrial food in the diet, during the first (A), second (B) and third (C) 10-day growth periods. Solid lines are plotted for instances where the relationship was found statistically significant. (D–F): Box plot of the residual body mass (g) in aviary chicks, separated by sex and dietary treatment, in the first (D), second (E) and third (F) 10-day growth periods. Boxes correspond to median, first and third quartile, and whiskers to 1.5 times the interquartile range outliers. **, $P < 0.01$.

*& Greig, 1985; Ronconi et al., 2014*). Foraging at sea is known to be highly competitive, with frequent agonistic interactions that favor the largest birds (*Garthe & Hüppop, 1998; Hudson & Furness, 1988*) and, in the case of the Lesser Black-backed Gull, generally males (*Camphuysen et al., 2015*). Terrestrial food sources may thus be more frequently exploited by the less competitive individuals or during periods of marine food shortage (*Navarro et al., 2010; Tew Kai et al., 2013; Tyson et al., 2015*), although competitive displacement is also observed in gulls foraging on garbage (*Greig, Coulson & Monaghan, 1986; Monaghan, 1980*). Indeed, terrestrial food sources are signaled in other *Larus* species as a sub-optimal alternative to marine food (*Annett & Pierotti, 1999; O'Hanlon, McGill & Nager, 2017*), and this suggestion is here supported by the fact that field chicks of both sexes raised on a predominantly marine diet attained a larger size, as well as a higher residual body mass from 10 days post-hatch onwards. Given that post-fledging survival rates in seabirds are positively related to fledgling size and body condition (*Lindström, 1999; Braasch, Schauroth & Becker, 2009; Bosman, Stienen & Lens, 2016*), a marine foraging strategy is therefore likely to yield higher reproductive success in our coastal population.
However, the relationships between diet composition and chick development in free-ranging populations may be confounded by factors underlying the variation in the proportion of marine food delivered to chicks. These factors may include parental sex, age, experience, timing of breeding, synchronization between partners or environmental effects (e.g., *Ramos et al., 2008*; *Camphuysen et al., 2015*). Given that terrestrial food provided *ad libitum* appeared as suitable as marine food for raising chicks in a controlled aviary environment, or even better considering differences in body size, trends in free ranging chicks might be due to differences in food quantity, rather than quality between diets. Indeed, if marine food is the preferred resource for chick provisioning, and lower-quality birds are outcompeted at sea and therefore forage more often on land, terrestrial diets may be associated with lower food provisioning rates. This suggestion however relies on the assumption that the terrestrial diet provided to chicks raised in the aviary held no significant differences in quality with that provided to free-ranging chicks. However, it cannot be discarded that, amongst terrestrial diets, differences in quality between the diverse items gulls consume may also translate into differences in chick development. Thus, chicken provided in the aviary is likely to present a more suitable nutrient profile than some terrestrial items consumed by gulls in the study population, such as Earthworms (see Table SA1). Finally, environmental factors potentially mediating the relationship between chick diet composition and chick development include variation in the relative availability of marine and terrestrial food sources, which can additionally affect the relative costs and benefits of different feeding strategies.

Integration of field sampling with aviary experiments in this study suggests that a terrestrial diet may lower reproductive success due to limitations in providing sufficient amounts of food to the chicks. This influences predictions on the effects of changes in marine food availability resulting from the announced reform of the Common Fisheries Policy (*European Parliament, 2008*), which aims at ending the practice of discarding by 2019. Based on the predominantly marine signature of food provisioned to Lesser Black-backed Gull chicks in our study population, and the behavioural and reproductive responses to earlier sporadic cases of discard reduction in various *Larus* species (*Oro, Bosch & Ruiz, 1995*; *Chapdelaine & Rail, 1997*; *Regehr & Montevecchi, 1997*; *Camphuysen, 2013*; *Tyson et al., 2015*), it can be expected that the planned disappearance (or at least reduction) of fishery discards will affect breeding success in coastal breeding gulls. The impact of such discard ban may depend on the past history of the colony. For instance, the study colony at Zeebrugge was mainly founded by immigrants originating from nearby coastal colonies along the Southern North Sea that likely exploited fishery discards (*Seys et al., 1998*), and can thus be expected to contain a large number of marine-specialist individuals. The extent to which local adaptation and social learning affect the cost of provisioning chicks with terrestrial food should be further assessed by studying the relationships here described in populations that are mostly reliant on terrestrial food sources (e.g., *Coulson & Coulson, 2008*; *Gyimesi et al., 2016*), where specialists in different terrestrial foraging modes may be found. Moreover, early diet could affect an individual's proficiency at exploiting a particular foraging niche through ontogenetic effects on physiological and morphological traits (*Oudman et al., 2016*), further

affecting the cost-to-benefit ratio of individual foraging strategies. Finally, more in-depth evaluation of fitness costs and benefits in early diets will require exploring the extent of differential investment of each member of a breeding couple, and integrating hidden costs such as contamination (*Arcos et al., 2002*; *Jaspers et al., 2006*; *Santos et al., 2017*).

## CONCLUSIONS

Variation in the marine vs. terrestrial composition of the diet of free-ranging chicks, driven by differences in parental feeding strategies, resulted in impaired chick growth where a larger terrestrial component was found. Similar patterns do not arise when terrestrial and marine diets are provided *ad libitum* to hand-raised chicks. We suggest that anthropogenic terrestrial diets may lower reproductive success due to limitations in food quantity, rather than quality.

## ACKNOWLEDGEMENTS

Ghent University, VOC Oostende and the Flanders Marine Institute (VLIZ) provided the infrastructure for the aviary experiment. We would like to thank field assistant Hans Matheve and laboratory technicians Angelica Alcantara, Katja Van Nieuland and Viki Vandomme, as well as the staff and volunteers of the VOC Oostende for their help. M.Sc. students Nelle Meyers and Eva Van Wassenhove provided assistance in the aviary experiment.

### Funding

This study was funded by Research Foundation–Flanders (FWO) grant G0E1614N to Wendt Müller and Luc Lens. Alejandro Sotillo is funded by Fundação para a Ciência e a Tecnologia grant PB/BD/113792/2015 (FCT, Ministry of Science, Technology and Higher Education, Portugal) in the framework of the Biology and Ecology of Global Change (BEGC) doctoral program. Jan M. Baert is funded by the FWO (grant 12R7619N). The funders had no role in study design, data collection and analysis, decision to publish, or preparation of the manuscript.

### Grant Disclosures

The following grant information was disclosed by the authors:
Research Foundation–Flanders (FWO): G0E1614N.
Fundação para a Ciência e a Tecnologia grant: PB/BD/113792/2015.
Biology and Ecology of Global Change (BEGC).
Research Foundation–Flanders (FWO): 12R7619N.

### Competing Interests

The authors declare there are no competing interests.

## Author Contributions

- Alejandro Sotillo conceived and designed the experiments, performed the experiments, analyzed the data, prepared figures and/or tables, authored or reviewed drafts of the paper, approved the final draft.
- Jan M. Baert authored or reviewed drafts of the paper, approved the final draft, guidance for data analysis.
- Wendt Müller authored or reviewed drafts of the paper, approved the final draft, design and execution of field work.
- Eric W.M. Stienen conceived and designed the experiments, authored or reviewed drafts of the paper, approved the final draft, design and execution of field work.
- Amadeu M.V.M. Soares conceived and designed the experiments, approved the final draft.
- Luc Lens conceived and designed the experiments, contributed reagents/materials/analysis tools, authored or reviewed drafts of the paper, approved the final draft.

## Animal Ethics

The following information was supplied relating to ethical approvals (i.e., approving body and any reference numbers):

The Ethical Committee of Ghent University approved this research (project EC number 2015-017).

## Field Study Permissions

The following information was supplied relating to field study approvals (i.e., approving body and any reference numbers):

The University of Antwerp (Ethical Committee for Animal Experiments) approved this research (EC2013-73).

## Data Availability

Data is available at Mendeley Data: Sotillo, Alejandro (2019), ''Chick measurements and stable isotope signatures'', Mendeley Data, v3

http://dx.doi.org/10.17632/p7b2cc9vbw.3.

## Supplemental Information

Supplemental information for this article can be found online at http://dx.doi.org/10.7717/peerj.7250#supplemental-information.

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
