# Peer review of "Recently-adopted foraging strategies constrain early chick development in a coastal breeding gull"

_PeerJ, doi:10.7717/peerj.7250_

## Round 0.1 · original submission · Minor Revisions

Both reviewers agree that this is a sound study requiring relatively minor clarifications before publication. I agree with this judgement. In addition to the reviewers’ comments, I have provided comments based on my own reading of the manuscript. You may treat these as if they were from a third reviewer, i.e., make appropriate changes if the comments are valid and provide a specific explanation if you decide not to make changes. I have also added an annotated pdf with minor grammatical corrections suggested by highlighting words or punctuation with an inserted comment to suggest an alternative. In some cases of repeated errors, I did not repeat the comment if I thought the issue would be obvious.

I apologize for the delay in responding. Several manuscripts became ready for an editorial decision almost simultaneously at a busy period.

Editor’s comments
L23. Would ‘nutritional value’ be more appropriate than ‘energetic value’ because protein or other nutrients might also as important as energy or more so?
L62. Rather than simply state that they are heterogeneous, wouldn’t it be better to indicate the variety of observed patterns to indicate the magnitude and type of variability seen between studies?
L73-76. I am not convinced by the argument that a higher fitness cost of terrestrial foraging is more likely to reflect a decline in fishery discards than a cultural shift toward a new foraging opportunity. If marine food availability declines, terrestrial food of lower availability or quality might become relatively more valuable. So there would be a historical decline in fitness but under current conditions, the two sources could be similar, for example if an ideal free distribution or similar pattern was established between the two sources.
L77ff. I don’t think that the knowledge gap you are attempting to address has been developed with sufficient clarity in the previous paragraphs preceding the statement of objectives. Please make clear what is already established and what remains less certain.
L99, L117. Was selection truly random, e.g., by a random number procedure, or do you mean ‘haphazardly’?
L129. Are newly hatched gull chicks able and willing to feed from a dish or do they feed better if food is offered to them by a parent or substitute?
L155. Scientific name(s) for the earthworms (and maybe even potatoes).
L192. Do you need to specify t and provide units for all variables?
L237. In contrast to the text and Table 2, Fig. 2d appears to imply that the difference between sexes was different (long bracket with 3 asterisks) while the difference between diets within sexes was not (no asterisks within sexes).
L237. According to Table 2, the p-value for a difference between the sexes was 0.08. Would it be accurate to state that there was a non-significant trend than simply that there was no difference?
L239ff. Please present the information in this paragraph parallel to the previous paragraph. Switching from contrasting field and aviary in each sentence in one paragraph to contrasting diet and sex effects in each sentence in the next paragraph creates unnecessary effort for readers to easily grasp the findings.
L242. Fig. 3b is confusing in a similar way to 2d, appearing to show an effect of diet and not sex.
L253. Fig. 4b is similarly confusing.
L264-265. Don’t the differences in asymptote and rate of body mass increase count as a clear difference?
L279. It is not clear what assumption you are referring to and whether it is an assumption, a hypothesis or a suggestion. If you are referring to the suggestion that less competitive individuals forage in the terrestrial environment, I think a more explicit argument is required to show the logical link between offspring size and competitiveness of parents. Assumption implies a step in a logical argument, but I’m not sure how it fits in here.
L283-284. ‘Most optimal’ is a redundant expression. Furthermore, terrestrial foraging might be optimal for individuals that cannot compete at sea. Why not be more explicit that it is likely to yield higher reproductive success?
L284. You need to be more explicit at how your data indicate that lower diversity is more successful? Presumably, it has to do with the negative slope of delta N, but you should be explicit in your assumption or evidence that lower levels of delta N represent higher diversity.
References: Please check the references for consistent formatting. Journal articles should not be in all caps. Scientific names should be italicized. Check whether you have used the correct abbreviation of J.F. Ornithol. The type of document for articles on L390 and 393 is unclear. Are these books, reports, journal articles, theses? Have your provided sufficient information that a reader could locate them?
Figures. It is a challenge to interpret data in some figures because of the sparse numbers on the axes (y and sometimes x). Please add tick marks and numbers at regular intervals over the whole range of values. Please check that all numbers on axes have the same level of precision. As I noted above, the brackets indicating significance are ambiguous. Please check that you are using the normal convention for positioning brackets, and explain the use of brackets in the captions. The conventional use of asterisks is * <0.05, ** <0.01, *** <0.001. It is not impossible to use other values, but it is likely to make the significance less clear to readers.
Fig. 2. Please add the units in the text and add the units for growth on the figure panels.
Tables. Most journals request that vertical lines be avoided in tables.
Table 1. Please add the word ‘proportion’ to the column head for marginal means.
Table 2 and 3. Will the meaning of ‘marginal mean/coefficient’ be clear to all readers? Please make sure that the units for this column in both tables are clear. Because of the change in measures between sections of Table 2, it might be possible to include them in column 2.
I was unable to check the supplemental isotope signatures file because I could not open it.

# Reviewer 1 ·

Basic reporting

The research questions / hypothesis are currently not well defined so please state these more explicitly.

The tables are currently quite busy and therefore not easy to read - please format these into a standardised formatted scientific table, ensuring columns and rows are well spaced.

Experimental design

The methods would benefit from a little more detail, for example, stating the dates colonies visits were made between, ensuring that it is clear that all free-ranging and aviary chicks are LBBGs, and explaining what the experimental nests are.

Validity of the findings

No comment

Additional comments

This is an interesting and concisely written manuscript exploring marine versus terrestrial diets in Lesser Black-backed Gulls. The experiment with aviary chicks is very useful given the variation in results from free-ranging chicks among colonies in different locations. The introduction provides concise and relevant background to the issues around establishing the consequences of switching from a predominantly marine to terrestrial diet in generalist gulls, and the discussion is concluded well with potential consequences of fishery closures, and how the results may differ if repeated in colonies which are more reliant of terrestrial resources.
1. My main comment is that some of the results do not seem to be discussed in the Discussion. For example aviary chicks reaching a higher asymptote body mass (and growth rate for body mass?) when raised on terrestrial food. It may also worth mentioning the higher body mass of males than females also found by other studies.
2. The research questions / hypothesis are currently not well defined so please state these explicitly.
3. The methods would benefit from a little more detail for example stating the dates colonies visits were made between, ensuring that it is clear that all free-ranging and aviary chicks are LBBGs, and explaining what the experimental nests are.

Line 27: I feel traditionally is a more appropriate word than originally
It would be useful to include a concluding line to the abstract to further highlight the importance of the result that the quantity of anthropogenic food items may lower reproductive success.
Line 54: Suggest replace growing with increasing
Line 68: Could this niche shift away from marine resources also be due to a decline in intertidal forging habitats due to increased coastal development and disturbance?
Line 94 and throughout: italicise latin species names
Line 96: change exhibits to exhibit
Line 102: Please provide the dates data was collected between, and were these visits made from egg laying until fledging? Please clarify.
Please state more explicitly in the methods that free-ranging and aviary chicks were Lesser Black-backed Gulls (i.e. not also Herring gulls).
Line 103: It is not currently clear what the experimental nests are? Please can you clarify. How many eggs did these experimental nests contain after receiving a first- or second-laid egg from other nests?
Line 122: How many eggs were collected in total from Ostend? It would be interesting to know how many of those collected hatched.
Line 150 – 153: refer to your Supplementary figures / tables.
Line 160: add for - cleaned for 5 minutes
Line 172 and throughout: please italicise δ and subscript the numbers i.e. δ13C
Line 173: Please include what species TEFs corrections were based on from Hobson and Clark (1992). i.e. Ring-billed Gull?
Line 176: is Stable Isotope Analysis a specific analysis within SIAR? Please expand on this.
It would be useful to know the number of male and female field and aviary chicks were sampled.
Line 265: Though aviary chicks reached a greater body mess when fed on a terrestrial diet? This would be worth discussing further.
Line 277: Although in some studies, larger male gulls have been found to exploit landfill resources to a greater extent than females thought to be due to larger ind. Outcompeting smaller individuals – see Monaghan, P. Anim. Behav., 1980, 28, 521-527.
Line 287: Be consistent in Capitlisation of species names – Herring Gull
Line 294: It could be based also on food quality given the range of different terrestrial items the gulls may consume and the differences in quality of these as shown by your Appendix Table 4 shows, which should be acknowledged.
Line 297-299: This is a good point as chicken is likely to contain more nutrients and protein than terrestrial items consumed in farmland such as grain and invertebrates, which is worth expanding on here given that your Lesser Black-backed Gulls consumed earthworms from pellet analysis.

Significant results in figures that are not discussed in discussion. i.e. with head length, body mass and chick condition in aviary chicks?
Figure 1. The palest grey for 20-30 days is not easy to see in plot b.
Figure 4. Does the black regression line just refer to males, and should there also be a line for females?
Table 1. The non-significant interaction between sex and age period was removed is repeated twice.
Table 3. Be consistent in how you abbreviate variables or write them out in full i.e. Prop. Terr. Food

Reviewer 2 ·

Basic reporting

In overall, the manuscript is well written, with clear objectives, nice introduction and discussion. Also, the tables and figures are enough to understand the results.

Experimental design

In my opinion, the experimental design is robust, allowing to answer the aims proposed by the authors.

Validity of the findings

The results are attractive to people working and this thematic.

Additional comments

Some minor comments to be addressed:

Lines 41: change “open-air landfills” to “landfills”. Basically because gulls also can exploit the organic matter present in non-open air landfills.

Line 48: Blonick et al 2003 and Araújo et al. 2011 are two references relating to the presence of individual specialization in diet.

Lines 51-53: I would like that the authors include some reference after this sentence

Lines 57 and 62: Some references that could be useful:
Mendes, R. F., Ramos, J. A., Paiva, V. H., Calado, J. G., Matos, D. M., & Ceia, F. R. (2018). Foraging strategies of a generalist seabird species, the yellow-legged gull, from GPS tracking and stable isotope analyses. Marine Biology, 165(10), 168.
Matos, D. M., Ramos, J. A., Calado, J. G., Ceia, F. R., Hey, J., Paiva, V. H., & Handling editor: Stephen Votier. (2018). How fishing intensity affects the spatial and trophic ecology of two gull species breeding in sympatry. ICES Journal of Marine Science, 75(6), 1949-1964.
Lines 63-65: See these two recent published papers:
van den Bosch, M., Baert, J. M., Müller, W., Lens, L., & Stienen, E. W. (2019). Specialization reduces foraging effort and improves breeding performance in a generalist bird. Behavioral Ecology.
Real, E., Oro, D., Martínez‐Abraín, A., Igual, J. M., Bertolero, A., Bosch, M., & Tavecchia, G. (2017). Predictable anthropogenic food subsidies, density‐dependence and socio‐economic factors influence breeding investment in a generalist seabird. Journal of Avian Biology, 48(11), 1462-1470.

Line 172: please, it is important to indicate the TEFs values and also how the authors estimated the TEFs.

Lines 175-178: MixSIAR model is more recommendable (more credible) than SIAR model. I suggest to estimate again the importance of terrestrial food in the chick’s diet by using MixSIAR modelling.

Line 158: Some reference regarding the feather growth rate is necessary in this sentence

Line 274-276: previously the authors remarked that this species shows low sexual dimorphism. For this, I do not understand why the authors discussed here that males could be better than females in relation to the exploitation of fishery discards.

Figures 2A, 2B, 3A and 3B: change “asymptote for body mass” to “body mass (g)”

---

## Round 0.2 · accepted · Accept

Thank you for the substantial and careful revision of your manuscript and your clear and detailed responses to the suggestions of the reviews and myself. I consider the manuscript now suitable for publication. There remain a few awkward choices of word use and a couple of minor issues. I will provide a pdf with highlights indicating the problem and comments to propose an improvement or explain the problem.

In addition:
L145. Should this ‘randomly’ also have been switched to ‘haphazardly’?
L264. I think that this distribution is skewed to the right in common usage, i.e. toward terrestrial diets. Check the Wikipedia page on ‘skewness’. However, I do seem to recall that there is some controversy about an opposite meaning.
References: There are still a few errors in the references, capital letters in journal titles, lower case letters in book titles, lack of italics in some genus and species names, capital letter for species names.
Figs. 2,3,4. Don’t the boxes represent the second and third, not first and third quartiles? Mention that outliers are indicated by filled circles. There is inconsistency between text, Tables 2 and 3, and Figures 2 - 4, sometimes referring to marine, fish or sprat and terrestrial or chicken. I suggest that you stick to the terms ‘terrestrial’ and ‘marine’ for the figures and keys. In the captions you can specify in parentheses for aviary studies that terrestrial food was ‘chicken’ and that marine food was ‘fish’. Note that this requires a change in the key in the figures.

#